# Relation of T Cell Profile with Vitamin D Receptor and Vitamin D-Binding Protein Gene Polymorphisms in Atopy

**DOI:** 10.3390/ijms25169021

**Published:** 2024-08-20

**Authors:** Daina Bastyte, Laura Tamasauskiene, Ieva Stakaitiene, Kamilija Briede, Rasa Ugenskiene, Skaidra Valiukeviciene, Brigita Gradauskiene

**Affiliations:** 1Department of Immunology and Allergology, Lithuanian University of Health Sciences, 50161 Kaunas, Lithuania; daina.bastyte@lsmu.lt (D.B.); laura.tamasauskiene@lsmu.lt (L.T.); 2Laboratory of Immunology, Department of Immunology and Allergology, Lithuanian University of Health Sciences, 50161 Kaunas, Lithuania; 3Department of Genetics and Molecular Medicine, Lithuanian University of Health Sciences, 50161 Kaunas, Lithuania; 4Department of Skin and Venereal Diseases, Lithuanian University of Health Sciences, 50161 Kaunas, Lithuania

**Keywords:** atopy, allergic asthma, atopic dermatitis, T cell profile, cytokines, VDR, GC, single nucleotide polymorphisms

## Abstract

Atopic diseases, including atopic dermatitis (AD) and allergic asthma (AA), are characterized by complex immune responses involving various T cells subsets and their cytokine profiles. It is assumed that single nucleotide polymorphisms (SNPs) in the Vitamin D receptor (*VDR*) gene and the Vitamin D-binding protein (*GC*) gene are related to the action of Vitamin D and, consequently, play a role in regulating the immune response. However, there is not enough data to unequivocally support the hypothesis about the relationship between T cells profile and *VDR* or *GC* SNPs. Two hundred sixty-six subjects (aged > 18 years) were involved in the study: 100 patients with mild or moderate AD, 85 patients with mild or moderate AA, and 81 healthy individuals. Blood cell counts were determined by standard methods. Flow cytometric analysis was used to evaluate CD4^+^ T-helper (Th) cell subtypes: Th2, Th1, Th17, and T regulatory (Treg) cells in peripheral blood. Measurements of cytokines, total immunoglobulin E (IgE), and Vitamin D levels in serum were evaluated by ELISA. Significantly higher levels of Th1, Th2, and Th17 cells, along with lower levels of Tregs, were found in patients with atopic diseases compared to healthy individuals. Additionally, higher serum levels of interleukin (IL) 5, IL-17A, and transforming growth factor-β1 (TGF-β1), as well as lower levels of IL-10, were observed in patients with atopic diseases than in control. The study established associations between *VDR* SNPs and immune profiles: the AA genotype of rs731236 was associated with increased Th2 and Th17 cells and a higher Th1/Th2 ratio; the GG genotype of rs731236 was linked to decreased serum IL-10 and TGF-β1 levels; and the TT genotype of rs11168293 was associated with increased IL-10 levels. Additionally, the GG genotype of *GC* gene SNP rs4588 was associated with reduced Th2 and Th17 lymphocytes, while the TT genotype of rs4588 was linked to decreased IL-10 levels. Furthermore, the CC genotype of rs7041 was associated with higher levels of Th2, Th17, IL-10, and IL-35, as well as reduced levels of TGF-β1, while the GG genotype of rs3733359 was associated with reduced IL-10 levels. In conclusion, our study demonstrates that the Vitamin D receptor gene single nucleotide polymorphisms rs731236 and rs11168293, along with polymorphisms in the Vitamin D-binding protein gene (rs4588, rs7041, rs3733359), are significantly associated with variations in T cell profiles in atopy. These variations may play a crucial role in promoting inflammation and provide insight into the genetic factors contributing to the pathogenesis of atopy.

## 1. Introduction

According to the latest publications by the European Academy of Allergy and Clinical Immunology (EAACI), atopic diseases are the most common chronic diseases in Europe, and their prevalence is expected to increase in the future [1]. These diseases affect approximately 20 percent of the world’s population [2], presenting new challenges for scientists and healthcare professionals [2].

Major atopic diseases such as atopic dermatitis (AD) and allergic asthma (AA) share the same atopic background and are closely associated with an imbalance in the number and function of CD4^+^ T-helper (Th) cell subsets Th2 and Th1 [3,4,5]. Although the development of atopy is primarily associated with increased Th2 cell activity, other T cell subtypes, such as Th17 and T regulatory (Treg) cells, also play significant roles in the pathogenesis of atopic diseases [3]. Additionally, the majority of studies show that in atopy, there is an imbalance between pro-inflammatory and anti-inflammatory immune cells [6,7,8,9]. It has been suggested that Th17-type cells and their associated cytokines may be actively involved in promoting inflammatory processes alongside Th2 cells, while Tregs and their cytokines may significantly contribute to suppressing inflammation [6,8,10]. Therefore, when evaluating the immune response to specify the nature of immune-response mechanisms during atopy, it is important to consider not only Th2 cells but also other T cell subtypes and their functional activity, including the cytokines secreted by these cells.

Moreover, atopic diseases are described as complex conditions influenced by a combination of environmental and genetic factors [5]. Environmental exposures, including allergens, pollutants, and lifestyle choices, play a crucial role in triggering and exacerbating allergic responses. Genetic predisposition also significantly contributes to the susceptibility and severity of these conditions related to atopy [5,11]. Among the factors involved in Vitamin D metabolism, alternative pathways of Vitamin D activation by CYP11A1 (cytochrome P450 side-chain cleavage enzyme), alternative nuclear receptors for Vitamin D hydroxyderivatives, and the Vitamin D receptor (VDR) and Vitamin D-binding protein (GC) have garnered attention for their roles in immune regulation [12,13,14,15]. Moreover, Vitamin D is known to modulate the activity of T cells, which are pivotal in allergic inflammation [12]. Single nucleotide polymorphisms (SNPs) in the *VDR* and the *GC* genes can affect the metabolism and function of Vitamin D, thereby influencing immune responses [15]. These polymorphisms may alter the balance of Th1, Th2, Th17, and Treg cells, which are critical in the pathogenesis of allergic diseases. Despite the growing evidence, more research is needed to fully elucidate the interactions between these genetic variations and environmental factors in the context of allergic diseases.

There is evidence suggesting that Vitamin D plays a significant role in the regulation of immune response in atopic diseases [12]. Studies have shown that Vitamin D contributes to the regulation of Th lymphocyte proliferation and differentiation, and that it affects cytokine synthesis processes [16,17,18,19]. Additionally, it is believed that serum Vitamin D levels and components involved in its metabolism, such as *VDR* and *GC* gene SNPs, may be related to the regulation of the immune response and the expression of atopic diseases [20].

Therefore, the aim of our study is to determine the relation of T cell profiles with Vitamin D receptor and Vitamin D-binding protein gene polymorphisms in subjects with atopy.

## 2. Results

### 2.1. Characteristics of the Studied Subject

The general characteristics of the study groups are summarized in Table 1. The studied groups did not differ significantly in terms of sex and age. Total immunoglobulin E (IgE) levels and blood eosinophil counts were significantly higher in subjects with AD or AA than in the control group (Table 1). Vitamin D levels were significantly lower in subjects with atopic diseases compared to the control group. Additionally, Vitamin D levels were higher in patients with AD than in those with AA (Table 1). There were no significant differences in total IgE levels and blood eosinophil counts between subjects with different atopic diseases (patients with AD and AA).

### 2.2. Distribution of Single Nucleotide Polymorphisms of VDR and GC Genes

When evaluating associations between *VDR* and *GC* gene SNPs and atopy, subjects were divided into atopic (*n* = 185) and control (*n* = 81) groups. The distribution of *VDR* gene polymorphisms did not show statistically significant differences between patients with atopic diseases and the control group (Table 2). In contrast, significantly more frequent detection of the *GC* gene SNP rs4588 genotype TT was found in the atopy group compared to the control group (Table 2). Rs4588 genotype TT was significantly more common in both atopic dermatitis and allergic asthma groups than in controls (13.0% and 14.6% vs. 3.8%, respectively; *p* < 0.05), with no significant difference observed between atopic dermatitis patients and those with allergic asthma. The distribution of other identified *GC* gene SNPs did not differ significantly among the groups, either when comparing the atopic group with the control group or between the atopic dermatitis, allergic asthma, and control groups.

### 2.3. T Cell Profiles

T cell profiles were determined in 86 subjects (61 with atopic diseases and 25 healthy subjects) (Table 3). Significantly higher levels of Th1, Th2, and Th17 cells, along with lower Tregs, were found in patients with atopic diseases compared to the control group (Table 3). Additionally, a higher ratio of Th17/Treg cells was observed in the atopic group compared to the control group (Table 3).

### 2.4. Levels of Pro-Inflammatory (IL-5, IL-17A, IL-33) and Anti-Inflammatory (TGF-β1, IL-10, IFN-γ, IL-35) Serum Cytokines

Serum levels of interleukin (IL) 5, IL-17A, IL-10, and transforming growth factor-β1 (TGF-β1) were determined in 175 subjects (141 with atopic disease and 34 healthy subjects), while serum levels of interferon-γ (IFN-γ), IL-35, and IL-33 were evaluated in 86 subjects (65 with atopic disease and 21 healthy individuals) (Table 4). The levels of Pro-inflammatory cytokines IL-5 and IL-17A were significantly higher in the groups of patients with atopic diseases compared to the control group (Table 4). Moreover, higher serum levels of TGF-β1 and lower of IL-10 were found in patients with atopic diseases compared to healthy individuals (Table 4). No significant differences were found between the AD and AA groups.

### 2.5. Association between T Cells, Cytokines, VDR, and GC SNPs in Atopy

Associations between T cells, cytokines, and *VDR* gene SNPs in atopy are shown in Figure 1 and Figure 2. The results revealed that the AA genotype of the *VDR* gene polymorphism rs731236 was significantly associated with higher levels of Th2 and Th17 cells, as well as a higher Th1/Th2 ratio, compared to the AG and GG genotypes of this polymorphism (Figure 1).

The GG genotype of rs731236 was significantly associated with lower levels of IL-10 compared to other genotypes of this polymorphism (Figure 2). Moreover, atopic subjects with the rs731236 GG genotype exhibited lower serum TGF-β1 levels than subjects with the GT genotype (Figure 2). Furthermore, higher levels of IL-10 were observed in subjects with the rs11168293 TT genotype compared to atopic subjects with other rs11168293 genotypes (Figure 2).

Associations between T cells, cytokines, and *GC* gene SNPs in atopy are shown in Figure 3 and Figure 4. It was found that the rs4588 GG genotype was associated with lower levels of Th2 cells compared to the GT genotype during atopy (Figure 3). Additionally, the rs4588 GG genotype showed significantly lower levels of Th17 cells compared to the TT genotype (Figure 3). Furthermore, the rs7041 CC genotype was associated with higher levels of both Th2 and Th17 cells compared to other genotypes of this polymorphism (Figure 3). Atopic subjects with the rs7041 CC genotype exhibited higher levels of Th17 cells than those with the CA or AA genotypes (Figure 3).

After evaluating the associations of *GC* gene polymorphisms with cytokines involved in the regulation of inflammation such as IL-5, IL-17A, IL-10, TGF-β1, IFN-γ, IL-35, and IL-33, the results showed that different genotypes of rs4588, rs7041, and rs3733359 were significantly associated with TGF-β1, IL-10, and IL-35 (Figure 4). The rs4588 TT genotype was associated with lower levels of IL-10 in atopy compared to the GT genotype (Figure 4). Additionally, atopic patients with the rs7041 CC genotype had lower levels of TGF-β1 and higher levels of IL-10 than those with the rs7041 AA genotype (Figure 4). The rs7041 CC genotype was associated with higher IL-35 levels compared to the AC genotype. Subjects with the rs3733359 GG genotype had significantly lower levels of IL-10 compared to those with the GT genotype of this polymorphism.

## 3. Discussion

This study revealed a significant relationship between T cell profiles and certain identified SNPs of *VDR* and *GC* genes, demonstrating a potential role of *VDR* and *GC* genes in the pathogenesis of atopy. As shown in other studies, genetic factors play a significant role in the development of atopic diseases through their influence on the immune response, whereas lifestyle and environmental factors are more related to the disease’s manifestations [21].

Evaluation of T cell subtypes in patients with atopic diseases and the control group showed significantly higher levels of Th2, Th1, and Th17 cells, and lower levels of Tregs in atopic patients compared to controls. Additionally, a higher Th17/Treg cells ratio was found in patients with atopy compared to the control group. Although no significant differences were found in the Th1/Th2 ratio, there was a tendency towards a higher ratio in atopic patients than in the control group. The evaluation of cytokines produced by T cells revealed significantly higher levels of IL-5 and IL-17A, lower levels of IL-10, and higher levels of TGF-β1 in the serum of atopic patients compared to the control group. Our results agree with studies that indicate the immune response in atopic diseases is associated with Th2 lymphocyte activity and an imbalance between Th1 and Th2 cells and the cytokines they produce [22,23]. There were no significant differences in Th1-related serum IFN-γ levels in patients with atopy compared to the control group. However, other studies have shown that cytokines synthesized by Th1 cells, especially IFN-γ, inhibit the production of Th2 cytokines [24,25]. Moreover, in a study by Vercelli et al., patients with atopic diseases were found to have increased levels of IL-4 released from active Th2 cells and, in response to active inflammation, significantly reduced IFN-γ production by Th1 cells [25]. In vitro studies have shown that IFN-γ inhibits IL-4-mediated IgE production by promoting the differentiation of naive T cells into Th1 cells [25]. In addition, another study shows that the treatment of atopic diseases such as AD with recombinant IFN-γ improves patient condition [26], although controversial results revealed that IFN-γ is not always associated with a reduction in serum IgE levels [27]. Moreover, this study is consistent with research demonstrating that atopic diseases are characterized by a relative lack of Tregs, which is associated with the activity of Th2 and Th17 cells [6,28,29]. Normally, Tregs suppress immune inflammation initiated by Th17 and Th2 cells by secreting cytokines such as TGF-β, IL-10, and IL-35 [30]. Additionally, naïve CD4+ T cells can differentiate into Tregs upon exposure to TGF-β and play an important role in regulating the number and functional activity of Th17 cells [28]. The increase in TGF-β1 found in the study may be related to the fact that TGF-β plays a significant role in promoting the differentiation of Tregs and suppressing the functional activity of Th17 cells in atopy [31,32]. This is particularly important in the regulation of immune inflammation, as the quantity and activity of Th17 cells can significantly influence the progression and severity of atopic diseases [10,28]. Moreover, these findings are consistent with studies indicating a correlation between the imbalance of Tregs and other CD4+ T cells and the development of atopic diseases [33]. Study results suggest that Tregs and their secreted cytokines, such as TGF-β1, regulate Th2 and Th17 cells, thereby inhibiting allergic inflammation. In addition, our results indicate that not only Th2 and Th1 cells but also Th17 and Tregs and the cytokines they produce, play crucial roles in the pathogenesis of atopic diseases.

Moreover, the study results indicated that patients with atopic diseases such as AD or AA had significantly lower levels of Vitamin D compared to the control group. It can be assumed that the reduced levels of Vitamin D in individuals with atopy are related to factors such as limited sun exposure, dietary habits, and genetic predispositions. Certain polymorphisms in genes involved in Vitamin D metabolism and immune function, such as *VDR* and *GC*, may increase susceptibility to both low Vitamin D levels and the development of atopic diseases [11,15]. Additionally, alternative pathways for Vitamin D activation, particularly those mediated by the cholesterol side-chain cleavage enzyme CYP11A1 (cytochrome P450 side-chain cleavage enzyme), may also influence Vitamin D levels [13]. Studies have shown that these alternative Vitamin D metabolites possess biological activity distinct from the canonical 1,25(OH)2D3 and are believed to act through different mechanisms and receptors [13,34]. However, the relationship between T cell profiles and factors such as Vitamin D or components involved in its metabolism, which may play a role in the pathogenesis of atopic diseases, is unclear. This suggests that Vitamin D may inhibit the differentiation of antigen-stimulated naïve CD4+ T cells into Th1 cells, suppress cytokine production in Th1 and Th17 cells, and regulate the development of Th2 and Treg cells [35,36]. Additionally, the discovery of new pathways for Vitamin D3 activation initiated by CYP11A1 [37] revealed that CYP11A1-derived Vitamin D metabolites, particularly 20(OH)D3 and 20,23(OH)2D3, act as inverse agonists of RORγ, suppressing its activity and leading to reduced IL-17 production [34]. Other studies confirm that Vitamin D contributes to regulating the balance between Th1 and Th2 lymphocytes, as well as modulating the immune balance between Th17 and Treg cells, and suggest that Vitamin D may contribute significantly to the suppression of allergic inflammation [38,39,40]. Moreover, the study by Looman et al. showed a positive association between 25(OH)D and effector memory T cells [41]. We did not analyze memory T cell levels in the studied patients. However, these results provide additional information about the role of Vitamin D in immune-response regulation and demonstrate the relationship of 1,25(OH)2D with both effector and memory T cells.

After evaluating the relationship between *VDR* SNPs and T cell subtypes, the results revealed that in patients with atopic diseases, the AA genotype of the *VDR* gene SNP rs731236 was associated with higher levels of Th2 and Th17 cells, as well as a higher Th1/Th2 ratio compared to other genotypes of this *VDR* SNP. These findings suggest that the AA genotype of rs731236 may be linked to more active inflammation in atopy. Furthermore, the GG genotype of rs731236 was significantly associated with IL-10 and TGF-β1 levels, which are indicative of Treg activity. This relationship may reflect a weakening of the Treg function, although no direct correlation between rs731236 and Treg levels was found. Additionally, there is evidence that the *VDR* SNP rs731236 may be associated with an increased risk of diseases such as asthma [42]. However, the association between *VDR* SNPs and T cell profiles, and the cytokines they express, is not yet fully understood. Our recent study revealed an association between *VDR* SNPs rs2228570, rs731236, rs11168293, and Vitamin D levels in atopy [43]. Moreover, VDR is an intranuclear protein found in various body cells, including bone, muscle, and immune system cells such as neutrophils, mast cells, macrophages, and activated T and B lymphocytes, as well as innate lymphoid cells [44,45]. The localization of VDR in immune cells suggests that the action of Vitamin D may be related to the activity of these cells. Vitamin D binding to VDR on mast cells is thought to induce the production of the cytokine IL-10, which inhibits the release of IL-1β, IL-2, IL-4, IL-6, TNF-α, and other cytokines, thereby regulating immune inflammation [46]. Additionally, mast cells are effector cells that play a critical role in the pathogenesis of IgE-dependent inflammation in atopy, so the action of Vitamin D via VDR on mast cells may influence the activity of Th2 and other CD4+ T cell subtypes [46]. Moreover, it can be assumed that not only polymorphisms of the *VDR* gene but also the expression of VDR in the cells were important for the obtained results [47]. Bendix et al. reported data from a study investigating the effect of Vitamin D treatment on VDR expression in T cells from patients with Crohn’s disease [47]. In Bendix et al.’s study, to assess cytoplasmic and nuclear VDR levels, mononuclear cells isolated from peripheral blood were stimulated with anti-CD3/CD28, stained using fluorochrome-labeled VDR antibodies, and evaluated by flow cytometry [47]. The results showed that the stimulated T cells produced a high amount of VDR, which was reduced by the action of active Vitamin D [47]. This demonstrates that Vitamin D can regulate T cell responses via VDR, thereby modulating pro-inflammatory mechanisms. However, there are conflicting studies showing that Vitamin D stimulates VDR expression in CD4+ T cells [48]. Nevertheless, Chang et al. showed that the treatment of mice with autoimmune encephalitis using an active VDR ligand and Vitamin D supplementation reduced the expression of IL-17 and alleviated disease symptoms [16]. The study also showed that in vitro treatment of CD4+ T cells with physiological doses of Vitamin D primarily inhibited Th17 cell cytokine production in a VDR-dependent manner without affecting the production of transcription factors or surface molecules [16].

The results of *GC* SNP analysis showed that the rs4588 GG genotype in individuals with atopy was associated with lower numbers of Th2 and Th17 cells compared to other genotypes of this polymorphism. Moreover, rs4588 TT genotype was associated with lower IL-10 levels compared to the GT genotype. These results suggest that the rs4588 GG genotype may play a protective role and significantly contribute to suppressing allergic inflammation in atopy. Meanwhile, the association of the rs7041 CC genotype with higher levels of Th2 and Th17 cells in peripheral blood and lower-serum TGF-β1 levels during atopy compared to other genotypes of this polymorphism reveals the possible role of this *GC* gene SNP in the activation of immune response and allergic inflammation during atopic diseases. Additionally, the rs7041 CC genotype was associated with higher levels of IL-10 and IL-35 compared to other genotypes of this polymorphism, suggesting that this *GC* gene SNP partially contributes to the regulation of Treg activity. The results also revealed significantly lower levels of IL-10 in subjects with the rs3733359 GG genotype compared to those with the GT genotype of this polymorphism. Based on data from other studies [49,50], it can be speculated that SNPs of the *GC* gene, a witch-encoding Vitamin D binding protein, may significantly regulate the immune system by inhibiting VDR signaling in immune cells, thereby limiting the interaction of Vitamin D with Th2, Th17, Treg, and other cells and controlling the entry of Vitamin D metabolites into cells.

*VDR* and different *VDR* SNPs may affect how effectively Vitamin D binds and acts in the body [51]. Additionally, GC protein and genetic variations in the *GC* gene may influence the binding and transport of Vitamin D [52,53]. Individuals with certain *VDR* or *GC* SNPs may require higher doses of Vitamin D or more frequent supplementation to achieve therapeutic effects. Personalized dosing, guided by genetic testing and Vitamin D levels, can help address issues related to Vitamin D binding and transport. Furthermore, since atopic diseases are often characterized by a Th2-dominated immune response and an imbalance between pro-inflammatory and anti-inflammatory mechanisms, using Vitamin D in combination with other immunomodulatory treatments may help regulate immune inflammation more effectively.

This study has several limitations. Statistical associations were evaluated, suggesting that further research on *VDR* and *GC* gene SNPs and the immune response in atopy is necessary. Although an association between specific *VDR* and *GC* gene SNPs, Vitamin D, and other components of the immune response in atopy has been identified, the causal relationships and their directions remain unclear. The identified associations should be considered as inferences that require more detailed exploration. Additionally, the association of *VDR* and *GC* genes with the pathogenesis of atopy and atopic diseases may be influenced by environmental factors, so the interaction between genes and environmental factors should not be ignored. The blood samples from the study subjects were collected between September and April. However, we did not include in the analysis all potentially related factors, such as the amount of solar radiation received, seasonal variations of Vitamin D, diet and eating habits, body weight and body mass index associations with the obtained results, and interactions of *VDR* and *GC* genes with other genes involved in Vitamin D metabolism, that may play an additional role. It is important to explore these interactions in future studies for a more complete understanding. Additionally, atopic diseases have dynamic and multifaceted characteristics. The present study provides an analysis of certain mechanisms involved in the pathogenesis of atopy, but the dynamic nature of atopic diseases requires continuous monitoring and research. Future studies should consider other research methods to assess the dynamics and etiology of atopic diseases. This study covers only Lithuanian residents, and the unique genetic, environmental, and cultural characteristics of this population may affect the generalizability of the results and the conclusions drawn. Therefore, the obtained data should be verified by conducting studies in other populations with different genetic and environmental contexts.

## 4. Materials and Methods

### 4.1. Study Design

A total of 266 subjects (aged > 18 years) were involved in the study: 100 patients with mild or moderate atopic dermatitis (AD) and 85 patients with mild or moderate allergic asthma (AA). The control group consisted of 81 subjects without known diseases that may influence study results. AD was diagnosed according to the criteria of Hanifin and Rajka [54], and subjects with AA were diagnosed and classified according to the Global Initiative for Asthma (GINA) guidelines [55]. Treatment of patients with AA includes inhalation of short- and long-acting adrenergic receptor agonists in combination with corticosteroids, while treatment for patients with AD involves topical corticosteroids and calcineurin inhibitors. The study was approved by the ethics committee (No. BE-2-74), and informed consent was acquired from all patients prior to sample collection. Blood samples were collected from September 2020 to April 2023 from subjects who did not take Vitamin D supplements. Exclusion criteria included any acute or chronic respiratory diseases (except asthma), pregnancy, and systemic autoimmune or oncological diseases.

To analyze the immune response during atopy, the amount of T cell subtypes—Th1, Th2, Th17, and Treg—in peripheral blood was evaluated in randomly selected 86 subjects (61 with atopic diseases and 25 subjects as control groups). To evaluate T cells’ functional activity, related cytokine levels were measured for some of the subjects: IL-5, IL-17A, IL-10, and TGF-β1 (141 with atopic disease and 34 healthy subjects), as well as IFN-γ, IL-35, and IL-33 (65 with atopic disease and 21 healthy subjects). *VDR* and *GC* SNPs, Vitamin D levels, total IgE, and blood eosinophil count were evaluated in all study participants.

### 4.2. Sample Collection and Storage

Peripheral venous blood samples were collected into K3 EDTA tubes for complete blood count and *VDR* and *GC* SNP analysis. Blood samples were also collected into lithium heparin tubes for evaluation of Th1, Th2, Th17, and Treg cells. Samples for evaluation of cytokines, total IgE, and Vitamin D assay were collected into serum tubes. Serum samples were stored at −80 °C for further evaluation.

### 4.3. Measurements of Total IgE, and Vitamin D

Measurements of total IgE levels in serum were performed using an ELISA commercial kit (IBL International, Hamburg, Germany). The limit of detection was 0.8 IU/mL. 

Vitamin D as serum 25-hydroxyvitamin D (25(OH)D) was evaluated by ELISA using a commercial kit (BioVendor, Brno, Czech Republic). Limit of detection was 2.81 ng/mL. 

### 4.4. Evaluation of VDR and GC Gene Single Nucleotide Polymorphisms

For single nucleotide analysis of *VDR* and *GC* genes, DNA was extracted from peripheral blood using the QIAamp DNA Blood Mini Kit (Qiagen, Hilden, Germany) according to the manufacturer’s instructions. *VDR* gene SNPs (rs7975232, rs1544410, rs731236, rs3847987, rs2228570, and rs11168293) and *GC* gene SNPs (rs4588 and rs7041) were analyzed using TaqMan single nucleotide genotyping assay probes (Thermo Fisher Scientific, San Jose, CA, USA) following the methodological instructions provided by the manufacturer.

### 4.5. Analysis of Peripheral Blood Cells and T Cell Profile

Peripheral blood cells were measured using an automated analyzer (Beckman Coulter, Miami, FL, USA).

For analysis of the T cell profile (Th1, Th2, Th17), peripheral blood mononuclear cells (PBMCs) were isolated from peripheral blood using gradient centrifugation (Ficoll Paque; Amersham Biosciences AB, Uppsala, Sweden) according to the manufacturer’s instructions. PBMCs were washed twice with Pharmingen Stain Buffer (FBS) (BD Biosciences, San Diego, CA, USA) and then suspended in RPMI 1640 (Corning, Corning, NY, USA) supplemented with 10% heat-inactivated fetal bovine serum and 100 units/mL of penicillin/streptomycin. The suspended cells were then stimulated with a Leukocyte Activation Cocktail, with GolgiPlug (BD Biosciences, East Rutherford, NJ, USA), for 4 h at 37 °C. After stimulation, the cells were washed twice with FBS and fixed with Cytofix™ Fixation Buffer (BD Biosciences, USA) for 15 min. Following fixation, the cells were washed twice and permeabilized with Perm/Wash™ Buffer (BD Biosciences) for 15 min, then stained for 30 min in the dark with anti-IgG isotype antibodies (BD Biosciences, USA) or a Human Th1/Th2/Th17 Phenotyping Kit cocktail (BD Biosciences, USA), consisting of fluorophore-conjugated anti-CD4-PerCP-Cy5.5, anti-IL-17A-PE, IFN-γ-FITC, and IL-4-APC.

For analysis of Tregs, 100 µL of fresh blood per tube were stained for 20 min in the dark with fluorophore-conjugated anti-CD4 and anti-CD25 antibodies (BD Biosciences, USA). After staining, erythrocytes were lysed, and the cells were washed with FBS and fixed with Human FoxP3 Buffer A (BD Biosciences, USA) for 10 min in the dark. After fixation, the cells were washed twice and permeabilized with Human FoxP3 Buffer C (BD Biosciences) for 30 min in the dark. The cells were then washed twice and stained for 30 min in the dark with a conjugated FoxP3 antibody (BD Biosciences).

The samples were analysed by flow cytometry with the BD FACLyrics system (BD Biosciences, USA), with 15,000–25,000 CD4 positive (CD4+) cells gathered in each experiment and lymphocytes gated based on the properties of their forward and side light scatter.

### 4.6. Measurements of Serum Cytokines

Measurements of Serum Cytokines IL-5, IFN-γ, IL-17A, IL-10, TGF-β1, IL-35, and IL-33 were performed by the enzyme-linked immunosorbent assay (ELISA) using commercial kits (Elabscience Biotechnology Inc., Huston, TX, USA). The sensitivity of the assay measurements was: IL-5—9.38 ng/mL, IFN-γ—9.38 ng/mL, IL-17A—18.75 ng/mL, IL-10—0.94 ng/mL, TGF-β1—0.1 ng/mL, IL-35—9.38 ng/mL, and IL-33—9.38 ng/mL.

### 4.7. Statistical Analysis

Statistical analysis was performed using IBM SPSS Statistics, Version 29 (IBM Corp., Armonk, NY, USA) and Microsoft Excel, Version 16 (Microsoft, Redmond, WA, USA). The methods of statistical analysis were selected after performance of Kolmogorov–Smirnov test. The significance of the differences between the groups of patients and control group was assessed using the Student’s *t*-test and one-way ANOVA. Due to the skewed distribution of the variables, nonparametric tests Mann–Whitney U or Kruskal–Wallis H were used. The proportions between the groups were assessed using Fisher’s exact test. All data were expressed as mean ± SEM. *p* values less than 0.05 were considered significant in all cases.

## 5. Conclusions

In conclusion, our study demonstrates that the Vitamin D receptor gene single nucleotide polymorphisms rs731236 and rs11168293, along with polymorphisms in the Vitamin D-binding protein gene (rs4588, rs7041, rs3733359), are significantly associated with variations in T cell profiles in atopy. These variations may play a crucial role in promoting inflammation and provide insight into the genetic factors contributing to the pathogenesis of atopy.

## Figures and Tables

**Figure 1 ijms-25-09021-f001:**
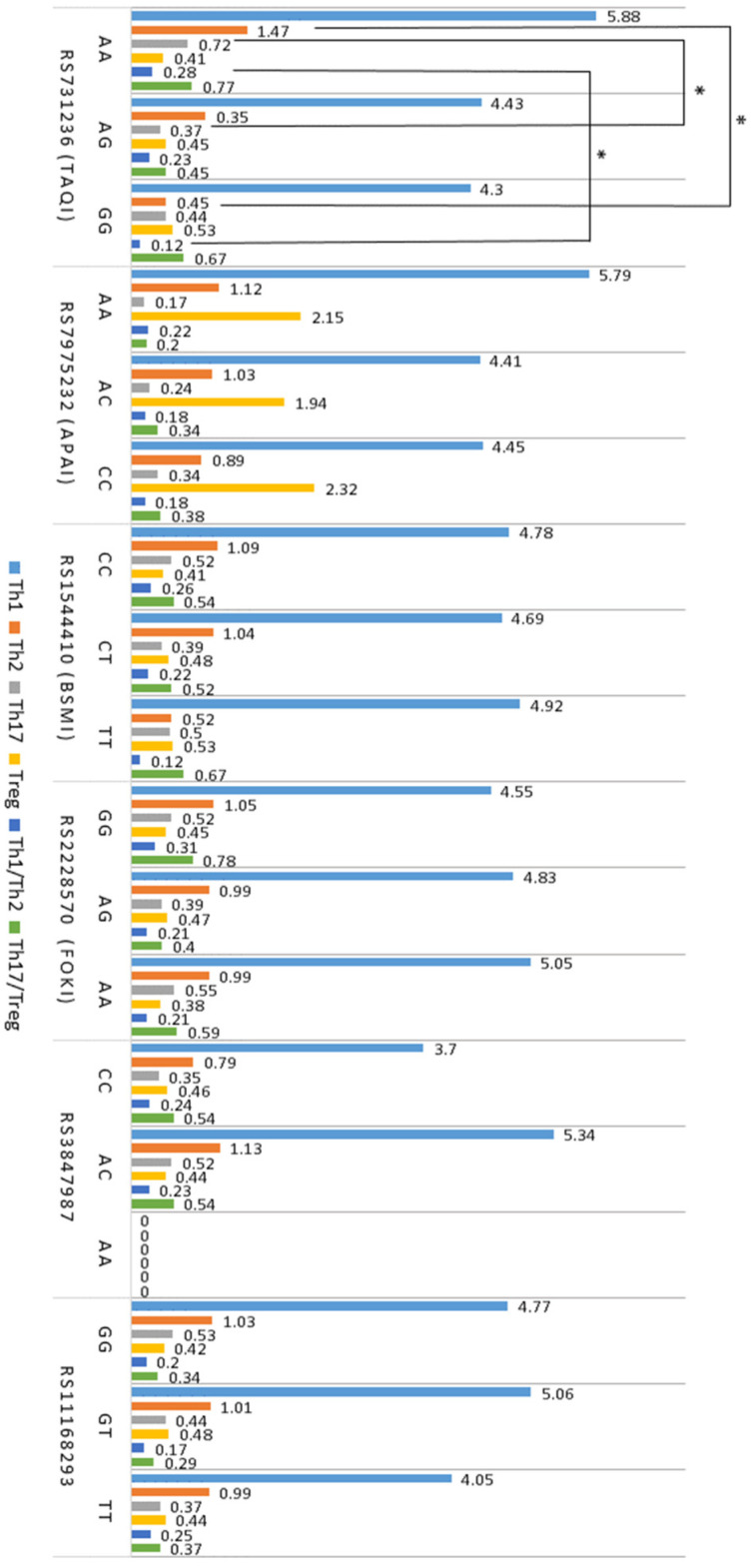
Associations between *VDR* SNPs and T cells in atopy (*n* = 61). Data are presented as the mean, with lymphocyte populations expressed as a percentage of PBMCs. * *p* < 0.05 when comparing the genotypes of *GC* gene polymorphisms.

**Figure 2 ijms-25-09021-f002:**
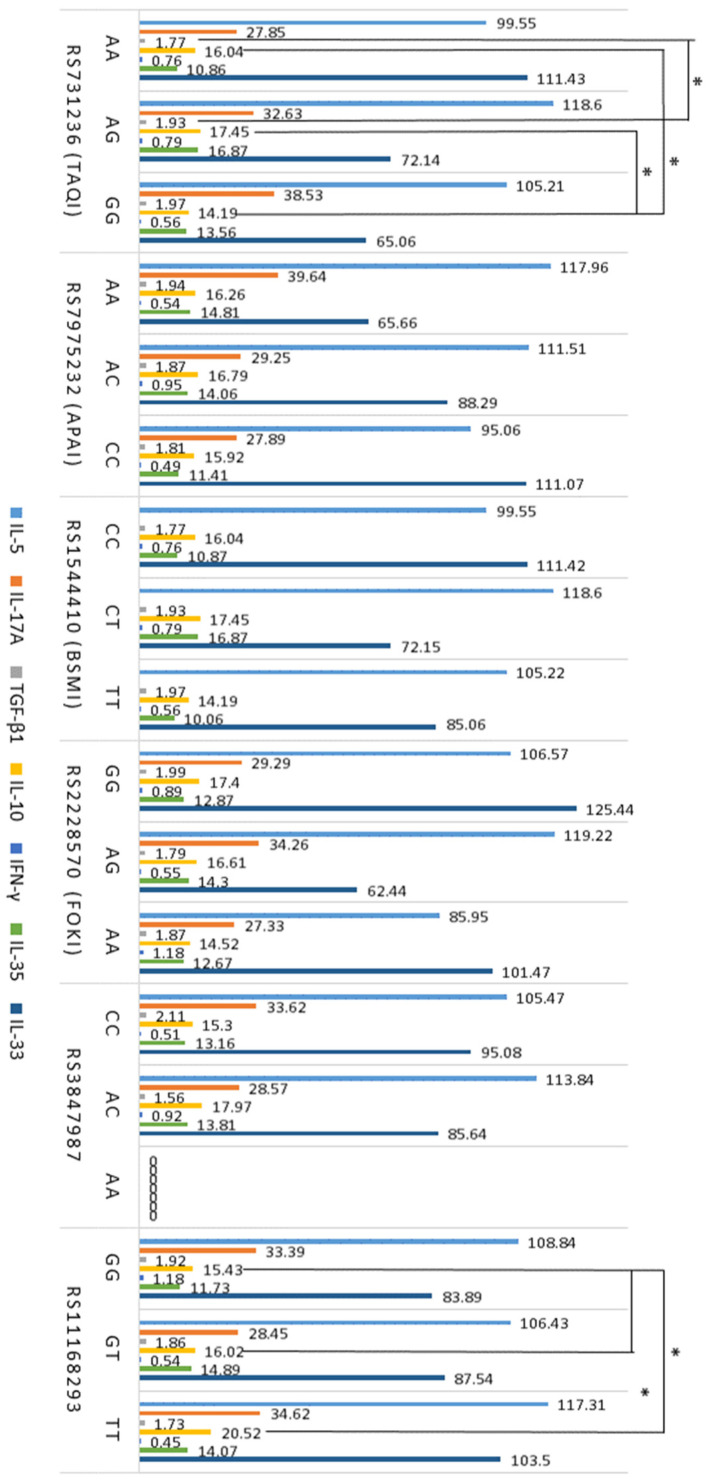
Associations between *VDR* SNPs and levels of cytokines IL-5, IL-17A, TGF-β1, IL-10 (*n* = 141) and IFN-γ, IL-35, and IL-33 (*n* = 65) in atopy. The data are presented as mean, with serum cytokine levels expressed in pg/mL, and TGF-β1 levels expressed in ng/mL. * *p* < 0.05 when comparing the genotypes of *GC* gene polymorphisms.

**Figure 3 ijms-25-09021-f003:**
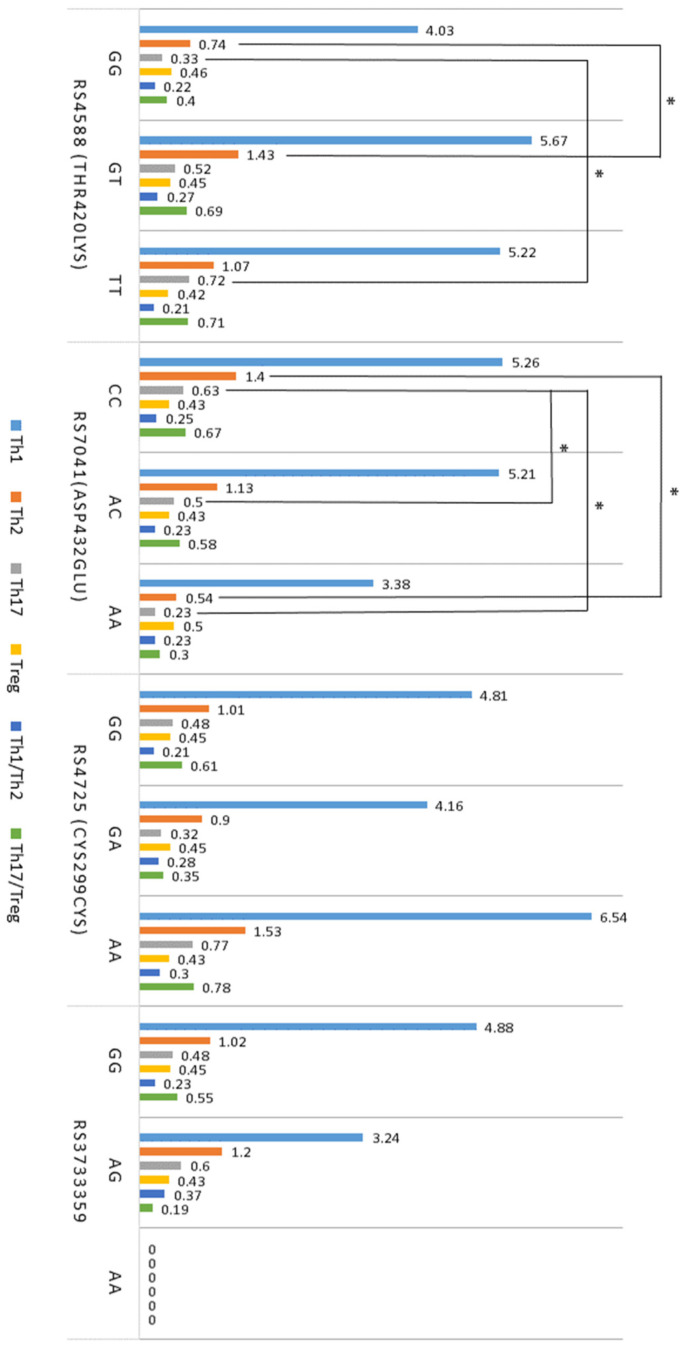
Associations between *GC* SNPs and T cells in atopy (*n* = 61). Data are presented as the mean, with lymphocyte populations expressed as a percentage of PBMCs. * *p* < 0.05 when comparing the genotypes of *GC* gene polymorphisms.

**Figure 4 ijms-25-09021-f004:**
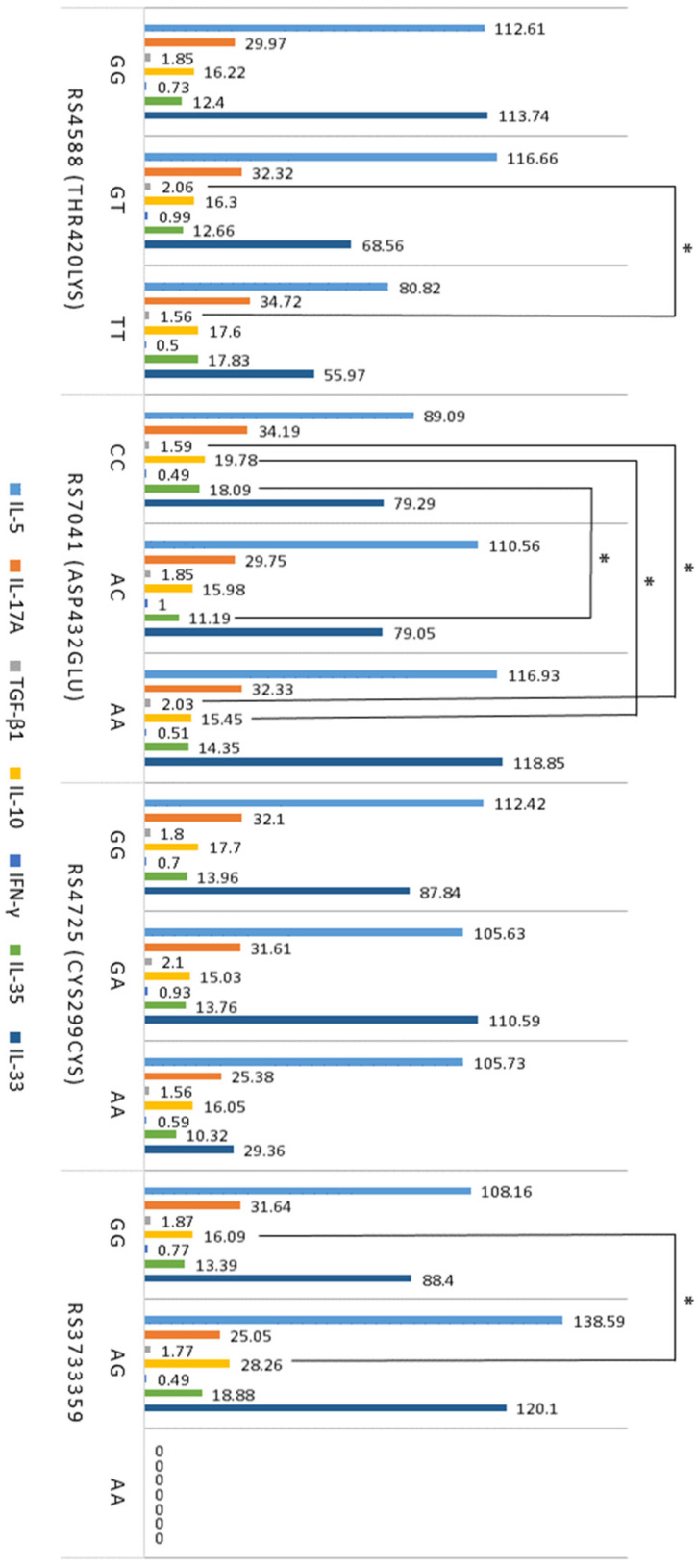
Associations between GC SNPs and levels of cytokines IL-5, IL-17A, TGF-β1, IL-10 (*n* = 141), and IFN-γ, IL-35, IL-33 (*n* = 65) in atopy. The data are presented as mean, with serum cytokine levels expressed in pg/mL and TGF-β1 levels expressed in ng/mL. * *p* < 0.05 when comparing the genotypes of *GC* gene polymorphisms.

**Table 1 ijms-25-09021-t001:** General characteristics of the study groups.

	Atopic Dermatitis (*n* = 100)	Allergic Asthma (*n* = 85)	Control Group (*n* = 81)
Male/female, n	31/69	30/54	32/49
Age (years)	29.51 ± 1	40.36 ± 1.5	35.57 ± 1.45
Total IgE, kU/L	919.83 ± 213 **	692.72 ± 170 *	24.99 ± 5.7
Blood eosinophils × 10^9^/L	0.40 ± 0.04 **	0.33 ± 0.06 **	0.06 ± 0.01
Vitamin D level, ng/mL	24.11 ± 0.94 *^ʃ^	18.37 ± 0.83 **	27.23 ± 1.21

Values are presented as mean ± SEM, * *p* < 0.05 compared with control; ** *p* < 0.001 compared with control; ^ʃ^
*p* <0.05 compared atopic dermatitis and asthma groups.

**Table 2 ijms-25-09021-t002:** Distribution of *VDR* and *GC* SNPs in patients with atopic diseases (*n* = 185) and healthy control subjects (*n* = 81).

Gene	SNP	Group	Genotype Frequency (%)			MAF
*VDR*	rs731236 (TaqI) A > G		AA	AG	GG	G
		Atopy	33.1	55.2	11.6	39.2
		Control	26.0	62.3	11.7	42.8
	rs7975232 (ApaI) A > C		AA	AC	CC	C
		Atopy	26.5	48.6	24.9	49.2
		Control	25.9	46.9	27.2	49.4
	rs1544410 (BsmI) C > T		CC	TC	TT	T
		Atopy	42.2	45.9	11.9	34.9
		Control	39.2	48.1	12.7	36.7
	rs2228570 (FokI) G > A		GG	GA	AA	A
		Atopy	31.4	50.3	18.4	43.4
		Control	27.2	53.1	19.8	46.2
	rs3847987 C > A		CC	CA	AA	A
		Atopy	61.1	36.2	2.7	20.8
		Control	55.6	43.2	1.2	22.6
	rs11168293 G > T		GG	GT	TT	T
		Atopy	42.7	40.5	16.8	37.1
		Control	42.0	43.2	14.8	36.4
*GC*	rs4588 (Thr420Lys) G > T		GG	GT	TT	T
		Atopy	52.7	33.5	13.7 *	30.5
		Control	53.2	43.0	3.8	25.3
	rs7041(Asp432Glu) C > A		CC	AC	AA	A
		Atopy	35.0	48.3	16.7	40.9
		Control	31.3	58.8	10.0	39.4
	rs4725 (Cys299Cys) G > A		GG	GA	AA	A
		Atopy	45.0	49.7	5.3	30.2
		Control	38.2	59.2	2.6	32.2
	rs3733359 G > A		GG	GA	AA	A
		Atopy	95.1	4.4	0.5	2.7
		Control	93.7	6.3	0	1.1

SNP—single nucleotid polymorphism; MAF—minor allele frequency. * *p* < 0.05 compared with control.

**Table 3 ijms-25-09021-t003:** Levels of T cell subtypes and their ratios in the studied groups.

T Cell Subset, % from PBMC	Atopic Dermatitis (*n* = 32)	Allergic Asthma (*n* = 29)	Control Group (*n* = 25)
Th1 (CD4^+^IFN-γ^+^)	5.78 ± 0.61 *	5.88 ± 0.82 *	3.39 ± 0.56
Th2 (CD4^+^IL-4^+^)	1.51 ± 0.25 **	1.60 ± 0.85 **	0.57 ± 0.09
Th17 (CD4^+^IL17-A^+^)	0.53 ± 0.25 *	0.58 ± 0.13 *	0.25 ± 0.04
Treg (CD4^+^CD25^+^FoxP3^+^)	0.48 ± 0.03 **	0.41 ± 0.02 **	0.65 ± 0.04
Th1/Th2 ratio	0.21 ± 0.03	0.26 ± 0.02	0.19 ± 0.02
Th17/Treg ratio	0.54 ± 0.06 *	0.65 ± 0.13 *	0.32 ± 0.03

PBMC—peripheral blood mononuclear cells. * *p* < 0.05 compared with control; ** *p* < 0.001 compared with control.

**Table 4 ijms-25-09021-t004:** Levels of pro-inflammatory (IL-5, IL-17A, IL-33) and anti-inflammatory (TGF-β1, IL-10, IFN-γ, IL-35) serum cytokines in the studied groups.

Serum Cytokine	Atopic Dermatitis	Allergic Asthma	Control Groupe
	*n* = 98	*n* = 43	*n* = 34
IL-5, pg/mL	110.8 ± 8.55 **	104.1 ± 12.2 **	46.9 ± 5.32
IL-17A, pg/mL	31.0 ± 2.81 *	32.82 ± 5.56 *	14.3 ± 1.34
IL-10, pg/mL	15.48 ± 0.87 **	19.28 ± 1.18 *	25.10 ± 2.41
TGF-β1, ng/mL	18.8 ± 0.07 *	18.2 ± 0.12 **	12.9 ± 0.09
	*n* = 36	*n* = 29	*n* = 21
IFN-γ, pg/mL	0.74 ± 0.19	0.81 ± 0.29	0.71 ± 0.22
IL-35, pg/mL	13.26 ± 1.56	15.18 ± 3.51	11.12 ± 0.44
IL-33, pg/ mL	103.15 ± 31.21	81.76 ± 16.20	36.21 ± 10.64

* *p* < 0.05 compared with control; ** *p* < 0.001 compared with control.

## Data Availability

All data generated or analyzed during the study are included in this published article.

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
