# Peer review of "Relation of T Cell Profile with Vitamin D Receptor and Vitamin D-Binding Protein Gene Polymorphisms in Atopy"

_ijms, 2024, doi:10.3390/ijms25169021_

Round 1

Reviewer 1 Report

Comments and Suggestions for Authors

The paper, in opinion of this reviewer is of interest.

The methodology and data presentation appear to be adequate. 

However, the final presentation is scientifically deficient because of following gaps in knowledge.

The authors do not consider or are not aware of alternative pathways of vitamin D activation by CYP11A1, and alternative nuclear receptors for vitamin D hydroxyderivatives (Biological Effects of CYP11A1-Derived Vitamin D and Lumisterol Metabolites in the Skin. J Invest Dermatol, 2024; doi:10.1016/j.jid.2024.04.022). This has to be corrected in the introduction and discussion. Make sure also to refer to original papers in addition to the above review where appropriate.

Note, that IL17 is downstream of RORg, and CYP11A1 derived D3-metabolites act on it as inverse agonists  (FASEB J 28:2775-2789, 2014).

Comments on the Quality of English Language

Minor proof-reading is advised

Reviewer 2 Report

Comments and Suggestions for Authors

This is an interesting study, which aim to determine the relation of T cells profile with

vitamin D receptor and vitamin D-binding protein gene polymorphisms in patients with

atopic diseases. I have following comments: 

1.     Please describe the treatment conditions of these recruited patients in details, and when the PB blood samples were collected for this study.

2.     Have you analyzed how SNPs in the VDR gene and GC gene affect memory T subsets, their phenotypic changes?

3.     As you mentioned, Vitamin D is also known to modulate metabolism. It’s interesting to identify how SNPs in the VDR gene and GC gene affect T cell metabolism, especially those elevated or decreased T cell populations. Have you ever isolated T subsets and analyzed metabolic changes instead of cytokines? 

4.     Do you have any evidence of how SNPs in the VDR gene and GC gene affect Vitamin D signaling pathways and downstream targets in T cells of atopic patients? 

5.     Can you discuss the anti-inflammatory role of INF-r in pathogenesis of atopic patients? 

6.     Can you discuss how to reverse such conditions to treat atopy? 

Round 2

Reviewer 1 Report

Comments and Suggestions for Authors

The authors were responsive to the critique and adequately revised the manuscript.

There are still typographical errors. For example, some titles in reference are capitalized and some not. Thes can be corrected at proofs stage.

Please pay attention to the detail.

Comments on the Quality of English Language

check for typographical errors